# LEARNING TO MINE APPROXIMATE NETWORK MOTIFS

## ABSTRACT

Frequent and structurally related subgraphs, also known as network motifs, are valuable features of many datasets. However, strong combinatorial bottlenecks have made it difficult to extract motifs and use them in learning tasks without strong constraints on the motif properties. In this work we propose a representation learning method based on learnable graph coarsening, `MotiFiesta` which is the first to be able to extract large and approximate motifs in a fully differentiable manner. We build benchmark datasets and evaluation metrics which test the ability our proposed and future models to capture different aspects of motif discovery where ground truth motifs are not known. Finally, explore the notion of exploiting learned motifs as an inductive bias in real-world datasets by showing competitive performance on motif-based featuresets with established real-world benchmark datasets against concurrent architectures.

## 1 INTRODUCTION

In many application domains, observing an over-represented substructure in a dataset is seen as evidence for its importance in network function. For example, early studies on network motifs enumerated all possible over-represented small subgraphs across various datasets and uncovered structures which explain the behaviour of real-world systems have been discovered such as the feed-forward loop in gene regulatory networks, and the bi-parallel motif in ecological food chains (Milo et al., 2002). More recently, motif libraries have shown strong utility in many machine learning contexts such as (Jin et al., 2020), classification (Zhang et al., 2020; Acosta-Mendoza et al., 2012; Thiede et al., 2021; Besta et al., 2022), representation learning Bevilacqua et al. (2021); Cotta et al. (2021); Rossi et al. (2020) and explainability (Perotti et al., 2022). Although exhaustively mining motifs is known to be NP-hard (Yu et al., 2020), motif utility has made the discovery task a key challenge in data mining for the past 30 years. The aim of this work is to formally expose the task of motif mining to machine learning models in an effort to enhance both the discovery of motifs and the representation power of learned models.

Any motif mining algorithm has to solve two computationally intensive steps: subgraph search and graph matching. The process of discovering a new occurrence of the motif involves a search over the set of subgraphs of a given dataset which yields a search space that grows exponentially with the number of nodes in the dataset as well as in the motif. Next, for a candidate motif and subgraph, a graph matching procedure is needed to determine whether the candidate can be included in the set of instances of the motif. Despite these barriers, many motif mining tools have been proposed Nijssen & Kok (2004); Yan & Han (2002); Wernicke (2006), all of which rely on simplifications of the task or *a priori* assumptions about the desired motifs. These simplifications include bounding motif size Alon et al. (2008), topology constraints Reinharz et al. (2018), and simplified subgraph matching criteria such as strict isomorphism, among others.

Besides those, an important limitation that is often overlooked is the variability inherent to many motif sets particularly in biological networks. Network datasets often represent dynamic or noisy processes and algorithms which limit their matching procedure to exact isomorphism will overlook a large set of possible motif candidates. Some tools have addressed this challenge, again with strong limitations such as REAFUM (Li & Wang, 2015) which allows for errors in node labelling, and RAM (Zhang & Yang, 2008)which tolerates a fixed number of edge deletions within a given

motif. All of these constraints limit the set of observable motifs and can lead to us missing important features of datasets. For this reason, we emphasize that our proposed methodology is built to support the discovery of *approximate* motifs.

Recent success of graph representation learning, particularly in an unsupervised context, presents an opportunity for circumventing some of these bottlenecks  (Karalias & Loukas, 2020). Namely, by allowing graph embedding models to leverage the statistical properties of a dataset, we can cast the search and matching problems to efficient operations such as real-valued vector distances. Of course, as is common with neural methods we sacrifice convergence and exactness guarantees in exchange for flexibility and speed. In this regard, there has been extensive work on problems related to motif mining using neural architecture such as subgraph counting  Teixeira et al. (2022); Chen et al. (2020); Liu et al. (2020) and graph matching  Li et al. (2019); Fey et al. (2020). To our knowledge neural motif mining problems for approximate subgraphs have yet to be proposed. Although related work has shown promise in similar motif mining tasks, by  (Oliver et al., 2022) in domain-specific and partially differentiable applications, and applied to the related problems of frequent subgraph mining,  (Ying et al., 2020), and discriminative subgraph mining  (Zhang et al., 2020). Finally, the composability of differentiable models allows motif mining to act as pre-training module orienting classifiers towards a robust and rich feature sets.  For these reasons, we believe there is a need to formally introduce the motif mining problem to the ML community by providing appropriate benchmarking settings and proposing relevant methodology.

## 1.1 CONTRIBUTIONS

In this work, we (1) formalize the notion of motif mining as a machine learning task and provide appropriate evaluation metrics as well as benchmarking datasets, (2) propose `MotiFiesta`, a fully differentiable model as a first solution to learnable motif mining which discovers new motifs in seconds for large datasets, and (3) we show that motif mining could also serve as an effective unsupervised pre-training routine and interpretable feature selector in real-world datasets.

## 2 TASK

### 2.1 NETWORK MOTIF DEFINITION

We start from the classical definition of motif which is a subgraph with a larger frequency than expected  (Milo et al., 2002). More formally, let $g = (V, \mathcal{E}, X)$ be a connected subgraph drawn from a graph dataset $\mathcal{G}$, where $V$ is a set of nodes, $\mathcal{E} \subseteq V \times V$ is a set of edges, and $X \in \mathbb{R}^{|V| \times d}$ is a feature matrix.

The **frequency** of subgraph $g$ is given by:

$$f(g, \mathcal{G}) = |\{h : h \simeq g \quad \forall \quad h \subset \mathcal{G}\}| \tag{1}$$

where we count the number of subgraphs $h$ isomorphic to $g$. The raw frequency of a motif leads to the task of frequent subgraph mining (Jiang et al., 2013) where we are interested in finding the set of subgraphs $g$ with maximal frequency. However to obtain significant motifs, the frequency must be normalized by the frequency of the subgraph in an null model that preserves the generic properties of the original network while ablating significantly enriched subgraphs  (Milo et al., 2002). The null graphs give us a baseline expectation of subgraph occurrence and therefore points us toward significantly enriched subgraphs.

Given a randomized (aka null) dataset $\tilde{\mathcal{G}}$, $g$ is considered a **motif** of $\mathcal{G}$ if

$$\frac{|f(g, \mathcal{G})|}{|f(g, \tilde{\mathcal{G}})|} > \alpha \tag{2}$$

is sufficiently large for some threshold $\alpha \in [0, 1]$.

An **approximate motif** follows the same definition but we simply replace the isomorphism condition with a graph similarity function such as a graph kernel (Vishwanathan et al., 2010; Kriege et al.,

2020) or edit distance (Riesen & Bunke, 2009), to allow non-isomorphic but similar subgraphs contribute to $f(g, \mathcal{G})$.

Evaluating $f$ exactly is exponential in the size of the search space and of the motifs and thus intractable for many real-world settings. Here, machine learning methods offer the advantage of capturing statistical properties of datasets and allowing us to cast the problem in terms of efficient vector distances and similarities. Translating the task of identifying motif occurrences to a machine learning context, we view the process as a node labeling.

## 3   MOTIFIESTA: MOTIF MINING MODEL

Here, we describe the proposed approximate motif mining algorithm, `MotiFiesta`. We saw in Section 2.1 that a motif is a subgraph that occurs with a sufficiently large normalized concentration. The challenge of motif mining is to search for the set all such subgraphs and their occurrences in a large dataset of graphs. We propose to tackle the structural similarity and concentration problems by training two neural layers: a subgraph embedder $\phi$ and a subgraph density estimator $\hat{f}$. Meanwhile, the search for subgraphs that fit the motif definition borrows the intuition that motifs can be composed of smaller adjacent motifs and therefore one can limit the search space by looking around a current set of motifs for larger ones  (Schreiber & Schwöbbermeyer, 2010; Oliver et al., 2022; Kashtan et al., 2004b). We bring this search strategy to differentiable setting by adapting the Edge-Pool graph coarsening layer  (Diehl, 2019). Our customized EdgePool layer is trained to coarsen subgraphs that fit the motif definition into a single node so that the neighbours of a coarsened node become candidates for larger motifs and further coarsening. To decide which subgraphs to coarsen, we rely on a graph similarity function $K$ that enables defining similar subgraphs. We learn a model $\phi$ that maps similar subgraphs to close points in a vector space - see section 3.2. Finally, we rely on a vector density estimation model $\hat{f}$ to compare the frequency of subgraphs in our dataset to the ones in decoy datasets - see Section 3.3. Our model then learns to group together subgraphs more frequent in our dataset. The fully end-to-end nature of the proposed method is chosen to take advantage of the scalability benefits of neural representation methods along with the ability to connect motif mining directly with supervised learning models. The execution of `MotiFiesta` is illustrated in **Figure 1**.

### 3.1   SUBGRAPH SEARCH THROUGH LEARNABLE GRAPH COARSENING

The main intuition behind `MotiFiesta`'s motif search strategy is to take advantage of the composability of motifs in a differentiable manner by adapting EdgePool  (Diehl, 2019). For an input graph, the EdgePool layer collapses pairs of nodes connected by an edge into a single node to obtain a coarsened graph. The layer collapses edges based on a learned probability and computes a new embedding that combines embeddings of collapsed nodes into a single node in the coarsened graph. As edge pooling layers are stacked, each node of upper layers contains information from larger and larger subgraphs in the original graph. Moreover, because all pooling events are carried out over edges, we guarantee that the subgraph being represented by each node is always connected which inherently satisfies the first criteria of motifs. The twist that we introduce is to cast the decision to collapse an edge when the corresponding subgraphs fit the motif definition.

As we need to ensure that embeddings assigned to each correspond to structural similarity, we need to keep track of the subgraph being encoded by each coarsened node. We define the spotlight of node $\mathrm{SL}(u)$, as the subgraph in the original graph whose nodes have been coarsened into $u$. As initialization, we define spotlights for the original input graph as $\forall u \in V, \mathrm{SL}(u) = \{u\}$. Then for each coarsening event, a pair of adjacent nodes is collapsed into a new node $w$ so we have $\mathrm{SL}(w) = \mathrm{SL}(u) \cup \mathrm{SL}(v)$. At any point in the coarsening, the spotlight maps a newly created node to a unique connected subgraph in the input. Figure 1 illustrates the notion of a spotlight.

### 3.2   SUBSTRUCTURE REPRESENTATION LOSS

The first loss term of our model ensures that similarities in node embeddings encode structural similarities of the subgraphs (spotlights) they represent. This step is necessary for ensuring that subsequent density estimates in the embedding space point to regions of high structural similarity,

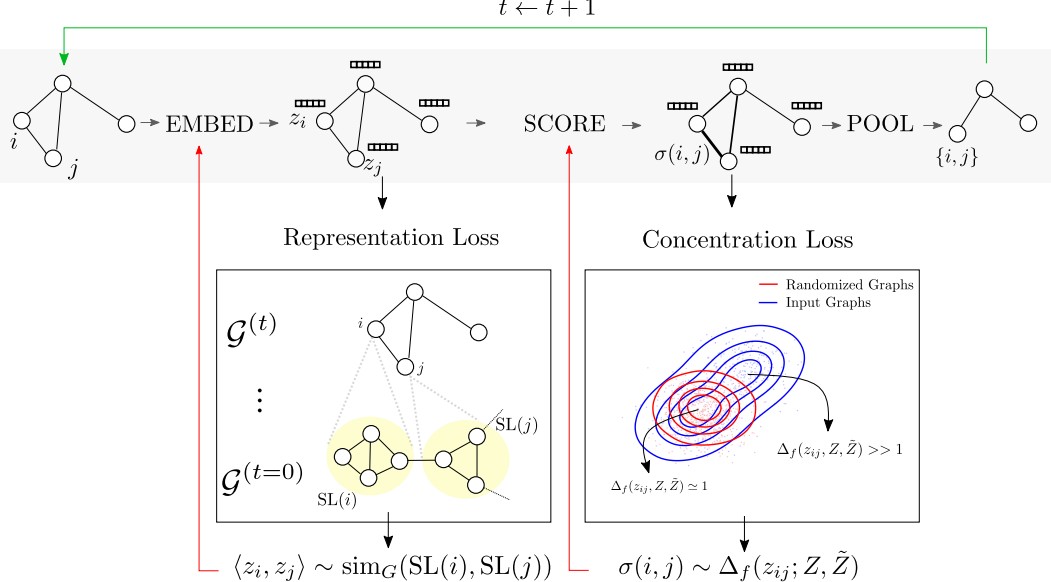

Figure 1: Execution flow of `MotiFiesta` through the view of one edge coarsening step $t$. Node embeddings $z_i$ are computed and trained to match a similarity function on the subgraphs of the original graph they encode. For each edge, a neural network assigns a coarsening probability $\sigma(i, j)$ to each edge according. The probability is trained to assign large values to embeddings with a large density in the data relative to a randomized graph dataset. Finally, we sweep through the edges of the graph while coarsening each with probability $\sigma(i, j)$ to create the graph at step $t + 1$

and allows us to easily accommodate approximate motif instances. Given any graph similarity function, $\text{sim}_G : (g_i, g_j) \rightarrow [0, 1]$, and a graph embedding function $\phi : g \rightarrow \mathbb{R}^d$, we define the *representation* loss for a pair of nodes $(u, v)$ as:

$$\mathcal{L}_{rep}(u, v, \phi) = ||\langle \phi(u), \phi(v) \rangle - \text{sim}_G(\text{SL}(u), \text{SL}(v))||_2^2. \quad (3)$$

To avoid a quadratic scaling factor with batch size $n$, we can randomly sample a fixed number of reference subgraphs $B$ to compute only $n \times B$ subgraph comparisons. The choice of similarity function will dictate the implicit rule determining motif belonging and thus application specific considerations can be encoded at this step (Oliver et al., 2022). Additionally, we are not restricted to functions with explicit feature maps or proper kernels and can use the inductive nature of $\phi$ to avoid evaluating similarity across all possible pairs of nodes. For these experiments we use the Wasserstein Weisfeiler Lehman graph kernel (Togninalli et al., 2019) which jointly models structural similarity and node feature agreement on general undirected graphs.

### 3.3 CONCENTRATION LOSS

The embedding model $\phi$ gives us node embeddings which represent connected subgraphs of the original graph. Next, we need to identify which of these subgraphs fits the concentration criterion of motifs. Let the embedding of a node $\phi(u)$ be $z_u$, remembering that the embedding is trained to represent the subgraph $\text{SL}(u)$. To assign a score to each edge at a given coarsening step $t$, we pre-compute coarsened embeddings $z_{uv}$ over all edges $(u, v) \in \mathcal{E}^{(t)}$, and train a scoring function $\sigma : z_{uv} \rightarrow [0, 1]$. The score is designed to assign a large probability to high density subgraphs. We implement $\sigma$ with a simple MLP and train it using an efficient vector density estimator $\hat{f}$ so that the second loss term is given by

$$\mathcal{L}_{concentration}(\sigma, \mathbf{Z}, \tilde{\mathbf{Z}}; \beta, \lambda) = -\sum_{i=1}^{|\mathbf{Z}|} \sigma(z_i) \exp[-\beta \Delta_{\hat{f}}(z_i; \mathbf{Z}, \tilde{\mathbf{Z}})] + \lambda \sum_i \sigma(z_i),$$

where $\Delta_f(z_i; Z, \tilde{Z}) = \hat{f}_Z(z_i) - \hat{f}_{\tilde{Z}}(z_i)$ is the difference in density estimates of $z_i$ under a batch of embeddings computed for the input graphs $\mathbf{Z}$ and randomized (rewired) graphs $\tilde{\mathbf{Z}}$, $\lambda$ is the regularization strength which shrinks the scores to zero, and $\beta$ controls the growth of $\sigma$ for more concentrated motifs. To generate $\tilde{Z}$ we apply the standard technique described in (Schreiber & Schwöbbermeyer, 2010) which iteratively swaps pairs of edges with each other such that local connectivity patterns are disrupted but the global graph statistics such as size and degree distribution are maintained.

Embeddings with a high density under the input graphs and low concentration under the randomized distribution will result in a large negative contribution to the loss. In this case a $\sigma(z_i) = 1$ will best minimize the loss. Non-concentrated subgraphs, or those with a high concentration in both distributions will have their $\sigma$ pushed to zero by the regularization term. As we expect motifs to occur less frequently than other embeddings, we apply an exponential to the delta function so that non-motif embeddings do not overwhelm the contribution of concentrated subgraphs.

There are several choices for the density estimate $\hat{f}$ but here we choose the $k$-NN density estimator (Mack & Rosenblatt, 1979) for its efficiency. This estimator is based on the intuition that a point in a dense region will on average have a small distance to its $k$-th nearest neighbor which can be efficiently computed using K-D trees. We therefore let $\hat{f}_{X,k}(x) = \frac{k}{N} \times \frac{1}{V^d R_k(x)}$ where $R_k(x)$ is the distance between $x$ and its $k$-th nearest neighbor, and $V^d$ is the volume of a unit $d$-dimensional sphere used for normalization.

A wrap up of these three components outlined in Algorithm 1 and model architecture details are in Appendix 6.3.

---

**Algorithm 1** `MotiFiesta` learnable edge contraction. We train a model to coarsen adjacent nodes if the subgraph they induce qualifies as a motif.

---

1: **Input:** a batch of graphs $\mathcal{G}$, graph similarity function $\text{sim}_G$, density estimator $\hat{f}$
2: **Output:** embedding model $\phi$ and scoring model $s$
3: $SL(u) \leftarrow \{u\}$ for each $u \in \mathcal{G}$ tracks subgraph of coarsened nodes
4: $\tilde{\mathcal{G}} \leftarrow \texttt{rewire}(\mathcal{G})$
5: $\mathcal{G}^{(0)} \leftarrow \mathcal{G}$
6: $\sigma, \phi \leftarrow$ initialize edge score and node embedding models
7: // Repeat this loop for $\tilde{\mathcal{G}}$
8: **for** $t = 1, \ldots, T$ **do**
9:    $\mathcal{G}^{(t)} \leftarrow \emptyset$
10:    **for** $(u, v) \in \mathcal{E}^{(t)}$ **do**
11:       $z_{uv} \leftarrow \phi(u, v)$ Joint embedding
12:       **if** $\sigma(z_{uv}) > \text{Uniform}(0, 1)$ **then**
13:          Add new node $w$ to $\mathcal{G}^{(t)}$ with feature vector $z_{uv}$
14:          Add edges from $w$ to $\text{Nei}(u)$ and $\text{Nei}(v)$.
15:          Update spotlights $\text{SL}(w) \leftarrow \text{SL}(u) \cup \text{SL}(v)$
16:       **end if**
17:    **end for**
18:    Add remaining unpooled nodes and edges from $\mathcal{G}^{(t-1)}$ to $\mathcal{G}^{(t)}$
19:    Backprop $\mathcal{L}_{rep}(\phi, z_u, z_v) \leftarrow ||\langle z_u, z_v \rangle - \text{sim}_G(\text{SL}(u), \text{SL}(v))||_2^2$ for all nodes.
20:    Backprop $\mathcal{L}_{conc}(\sigma, z_{uv}) \leftarrow -\sigma(z_{uv}) \cdot \exp\left[-\hat{f}_{\mathbf{Z}}(z_{uv}) - \hat{f}_{\tilde{\mathbf{Z}}}(z_{uv})\right]$ for all edges.
21: **end for**

---

## 3.4 DECODING

The output required to evaluate M-Jaccard coefficient is an assignment of nodes to a fixed number of discrete categories (motifs). However, the model does not assign embeddings to any categories, instead it only models the motif-likeness as a continuous property of all subgraphs in the dataset through the pooling score $\sigma$. To collapse this score to $K$ motifs we apply a locality sensitive hash (Dasgupta et al., 2011) to all the embeddings in the dataset. Each embedding (spotlight) is assigned to a bucket by a locality sensitive hashing function such that similar embeddings are assigned to the same bucket, leveraging the approximate motif properties enforced by our representation loss.

The bucket identifier can then be taken as an integer code of the nodes in the original graph. For each bucket, we are given the mean $\sigma$ score and keep only the top $K$ scoring buckets while the rest are assigned to the 'no motif' label. At this point we apply the M-Jaccard coefficient permutation test on datasets where have a ground truth motif labeling. We summarize the decoding execution in Algorithm 3.

## 3.5 COMPLEXITY ANALYSIS

The bulk of runtime is spent on the training step which is performed once per dataset. Since we chose the Wasserstein Weisfeiler Lehman graph kernel, evaluating the representation loss for $n$ nodes requires $\mathcal{O}(m^2)$ calls to the WWL kernel with runtime $\mathcal{O}(n^3 \log(n))$ where $m$ is the number of graphs in one batch. As an implementation detail, we were able to achieve good performance by computing a constant number of pairwise subgraph comparisons regardless of batch size to avoid the quadratic cost. The density estimation is built on a nearest neighbor estimator which relies on quick neighbourhood searches which can be implemented on a K-d tree in $\mathcal{O}(\log(m))$. In the induction phase, we execute a forward pass through the scoring and pooling which is $\mathcal{O}(m)$. To discretize the subgraphs we use an $\mathcal{O}(n)$ locality sensitive hashing procedure. We note that polynomial and sub-polynomial runtimes for classical enumeration methods are very rare. Typical training times on were around 2 hours for the synthetic datasets on a single GPU, and the decoding step takes $\sim 10$ seconds.

## 4 RESULTS

We test the proposed model and model evaluation framework in three settings. The first experiment tests the ability of `MotiFiesta` to retrieve motifs in synthetic datasets when the ground truth motifs are known. Finally, we explore the potential for the motif mining task as an unsupervised pre-training step in graph classification setting. Finally, to explore the relevance and interpretability of the motif mining procedure we perform an ablation study on our mined motifs to search for important motifs.

### 4.1 MINING FOR PLANTED MOTIFS

To evaluate the motif mining capacity of our trained models, we propose as in (Ying et al., 2020) to build a synthetic dataset where motifs are artificially injected at high concentrations at known positions (see Appendix 6.1). In this context, and in real-world datasets where ground-truth motifs are known, evaluating the performance of a motif miner can be done through efficient set comparisons through a small modification of the Jaccard measure.

We denote the (soft) labeling with $\hat{\mathbf{Y}} \in [0,1]^{|V| \times K}$ which specifies the probability that node $i$ belongs to motif $j$ for all graphs in a dataset where $K$ is the number of motifs in the dataset. In the synthetic setting, the number of motifs $K$ is known *a priori* but is not given to the model at training time. Note that this framing allows for soft (probabilistic) assignment of nodes to motifs, non-isomorphic motif occurrences, as well as one node belonging to multiple motifs (non-disjoint motifs).

When considering a single dimension (motif), we see that $\hat{\mathbf{Y}}_{*,j}$ partitions $\mathcal{G}$ into those nodes inside motif $j$, $\hat{\mathbf{Y}}_{*,j} \to 1$ and those outside the motif $\hat{\mathbf{Y}}_{*,j} \to 0$. At this point, we wish to measure the agreement between two sets for all motifs, for which the Jaccard coefficient is widely accepted. Here, we use the generalization of the Jaccard coefficient to real-valued sets. To evaluate the model applied to a single graph, we have

$$\mathbf{J}(\hat{\mathbf{Y}}, \mathbf{Y}) = \frac{1}{K} \sum_{j=1}^{K} \frac{\sum_{i=1}^{|V|} \min(\mathbf{Y}_{i,j}, \hat{\mathbf{Y}}_{i,j})}{\sum_{i=1}^{|V|} \max(\mathbf{Y}_{ij}, \hat{\mathbf{Y}}_{ij})}, \text{where} \quad \hat{\mathbf{Y}}, \mathbf{Y} \in [0,1]^{|V| \times K}. \quad (4)$$

The Jaccard coefficient ranges from 0 to 1 and simultaneously captures precision and recall. Models which over-assign nodes to a given motif are penalized by the denominator which considers the

union of the prediction and the true motif, while a model that misses motif nodes will have a low numerator value.

Because there is no inherent ordering to motif sets, the labeling assigned by the model and the one chosen for ground-truth motifs will be arbitrary. The final performance measure for a motif labeling is therefore given by the maximum Jaccard coefficient over all permutations $S_K$ of the columns of $\mathbf{Y}$:

$$\text{M-Jaccard}(\mathbf{Y}, \hat{\mathbf{Y}}) = \max_{\pi \in S_K} \text{J}(\hat{\mathbf{Y}}, \pi(\mathbf{Y})). \tag{5}$$

We generate several synthetic datasets that capture different motif-related variables. Each dataset consists of 1000 graphs generated randomly using the Erdös-Reyni random graph generator. Next, we generate one or more subgraphs that will act as the motifs. To create concentrated subgraphs we insert the motifs graphs into the each of the original graph and randomly connect nodes in the motif to the rest of the graph. In this manner we know which nodes belong to motifs and can control the concentration, size, topology, and structural variability of the motifs. This comes with a caveat since it is possible that the base graphs already contain motif occurrences naturally which would not be annotated as motif instances. It is possible to check the extent of this problem for simple motifs that have specified counting algorithms (e.g. cliques) which we show in Appendix 6. To mitigate this effect, we focus our evaluation on larger motifs (10 nodes) for most experiments.

We test three main motif mining conditions: motif topology, number of motifs in the dataset, and varying true concentration of the motif using the M-Jaccard. Results are summarized in Table 1 (continued in Appendix 8) and a sample of top scoring motif sets for various settings are shown in Figure 2. As a control condition, for each dataset we train model where all edge scores are fixed to 0.5. Before performing the M-Jaccard, we performed a simple check on the behaviour of the $\sigma$ scores to verify that motif nodes are assigned larger scores (see Appendix 7).

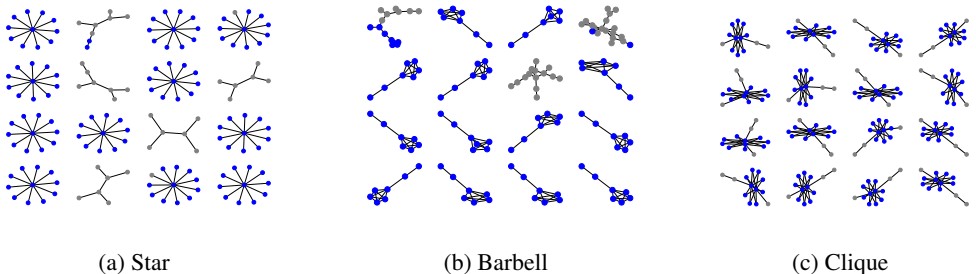

|          (a) Star          |          (b) Barbell          |          (c) Clique          |

Figure 2: Random samples from most populated buckets with ground truth motif nodes in blue.

The hyperparameter set remains the same for all experiments (see Appendix 6.3). Results for all conditions tested are summarized in Appendix. The decoding step uses a small grid search over the flexibility parameter in the hashing step (hash size) as well as in the pooling layer to use (subgraph size), we output the max average accuracy over 5 repetitions for each condition. `MotiFiesta` is able to consistently outperform the random baseline in many conditions. Notably, it appears that when the topology of the motif is well defined (as in the case of the barbell and clique) we see the strongest results, indicating that the power of the graph embedding model is an important factor. As external comparisons, we apply the classical Mfinder (Kashtan et al., 2004a) and neural method SP-Miner (Ying et al., 2020) on our synthetic motifs. Since Mfinder is limited to small isomorphic motifs we only test the different topologies and find that for all except the clique, we are able to achieve significantly higher performance (see Appendix 6.6). We find similar results when comparing to SP-Miner which is designed to mine the globally most frequent isomorphic subgraphs (see Appendix 6.7). We note that both these algorithms, while performing similar tasks, are not directly comparable to our *approximate and significant* motif mining setting.

## 4.2 MOTIF MINING AS FEATURE EXTRACTION

Apart from applications in data mining, we explore the potential for the motif task as a pre-training phase in supervised learning. We therefore test the view that motifs are statistically robust and

| | $\epsilon = 0$ | $\epsilon = 0.01$ | $\epsilon = 0.02$ | $\epsilon = 0.05$ |
|---|---|---|---|---|
| **barbell** | $0.68 \pm 0.04$ ($0.40 \pm 0.01$) | $0.66 \pm 0.05$ ($0.37 \pm 0.02$) | $\mathbf{0.69 \pm 0.01}$ ($0.32 \pm 0.05$) | $0.67 \pm 0.02$ ($0.34 \pm 0.06$) |
| **clique** | $0.58 \pm 0.14$ ($0.35 \pm 0.00$) | $\mathbf{0.62 \pm 0.08}$ ($0.34 \pm 0.00$) | $0.57 \pm 0.15$ ($0.34 \pm 0.00$) | $0.41 \pm 0.08$ ($0.34 \pm 0.00$) |
| **random** | $\mathbf{0.51 \pm 0.03}$ ($0.34 \pm 0.00$) | $0.49 \pm 0.02$ ($0.35 \pm 0.00$) | $0.46 \pm 0.02$ ($0.35 \pm 0.00$) | $0.44 \pm 0.04$ ($0.35 \pm 0.00$) |
| **star** | $0.43 \pm 0.09$ ($0.43 \pm 0.03$) | $0.42 \pm 0.02$ ($0.40 \pm 0.04$) | $0.40 \pm 0.02$ ($0.38 \pm 0.04$) | $0.40 \pm 0.03$ ($0.38 \pm 0.03$) |
| **5 nodes** | $0.37 \pm 0.02$ ($0.34 \pm 0.00$) | $0.36 \pm 0.00$ ($0.35 \pm 0.01$) | $0.36 \pm 0.01$ ($0.36 \pm 0.00$) | $0.36 \pm 0.00$ ($0.35 \pm 0.02$) |
| **10 nodes** | $\mathbf{0.48 \pm 0.02}$ ($0.35 \pm 0.00$) | $0.46 \pm 0.03$ ($0.35 \pm 0.00$) | $0.43 \pm 0.01$ ($0.38 \pm 0.04$) | $0.47 \pm 0.03$ ($0.35 \pm 0.01$) |
| **20 nodes** | $\mathbf{0.48 \pm 0.12}$ ($0.34 \pm 0.01$) | $0.43 \pm 0.08$ ($0.35 \pm 0.01$) | $0.44 \pm 0.03$ ($0.39 \pm 0.05$) | $0.33 \pm 0.00$ ($0.34 \pm 0.00$) |

Table 1: M-Jaccard score under various synthetic motif mining conditions. Each row contains the M-Jaccard score when testing on motif datasets with each edge of the motifs distorted with probability $\epsilon$. Unless specified otherwise, motif instances are of size 10 nodes. The values in small type and parentheses are the scores obtained by a dummy model that assigns all edges a merging probability of $0.5$. The largest score in each experiment is in bold when significant.

structurally complex feature sets. Taking three real world datasets in the TUDataset (Morris et al., 2020) from various domains we train `MotiFiesta` using only the representation and concentration loss again with rewired graphs as a negative distribution. Once the motif mining model is trained we compute a whole graph embedding by applying a global pooling to each merging layer and concatenate the output of each layer to produce a graph embedding so that for $T$ pooling layers of $d$ dimensions each, we obtain a $T \times d$ feature vector for each graph. Once training converges, we freeze `MotiFiesta` and pass the embeddings to a random forest classifier. In this manner, the embeddings are computed without any information about the classification labels. As baselines we train three graph neural network models in an end to end fashion (EdgePool, GCN (Kipf & Welling, 2016), GIN (Xu et al., 2018)). We report the results in **Table 2**.

| | PROTEINS | COX2 | IMDB-BINARY |
|---|---|---|---|
| **MotiFiesta** | $73.1 \pm 2.0$ | $80.7 \pm 2.4$ | $72.2 \pm 3.3$ |
| **EdgePool** Diehl (2019) | $73.6 \pm 4.1$ | $80.5 \pm 4.0$ | $71.8 \pm 3.6$ |
| **GCN** Kipf & Welling (2016) | $73.5 \pm 5.6$ | $80.9 \pm 4.0$ | $72.8 \pm 3.1$ |
| **GIN** Xu et al. (2018) | $72.6 \pm 3.9$ | $79.6 \pm 5.1$ | $73.4 \pm 3.2$ |

Table 2: Classification performance of `MotiFiesta` on graph classification benchmarks.

Across three different datasets, using motif-based embeddings has similar performance to state of the art graph neural network architectures. It cannot be said that this method improves the state of the art but given the unsupervised nature of `MotiFiesta`, this result suggests that computing embeddings solely by focusing the attention of the model towards statistically enriched subgraphs (motifs) can be a powerful and efficient way of extracting useful subgraph features. This provides support for the notion that frequent subgraphs are a useful inductive bias in prediction tasks.

### 4.3 MOST SIGNIFICANT MOTIFS CORRELATE WITH MODEL PERFORMANCE

In real-world datasets, we lack ground-truth motifs, making definite motif mining evaluation difficult. Despite this limitation, we propose to study the importance of the mined motifs through ablation studies on classification tasks. If the motifs identified in real-world datasets hold important information about the functioning of the network, excluding their embeddings from the global graph embedding should have an impact on classification performance. Next, we apply the same models trained for 2 but when computing the global graph embedding, we only allow subgraphs with the top $L$ $\sigma$ scores to contribute to the global graph embedding. The intuition is that the subgraphs which the model assigns the highest motif likeness should lead to the strongest classification performance. To control for the differing amount of information entering the global embedding with varying $L$, we also filter the embeddings by picking $L$ spotlights at random (red line). We see in Figure 3 a general upward trend in performance as $L$ grows while the top $L$ filtered models outperform the random model for low values of $L$. This indicates that in a fully unsupervised manner, the model is able to identify useful subgraphs.

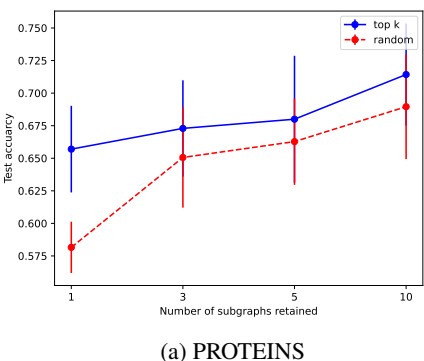
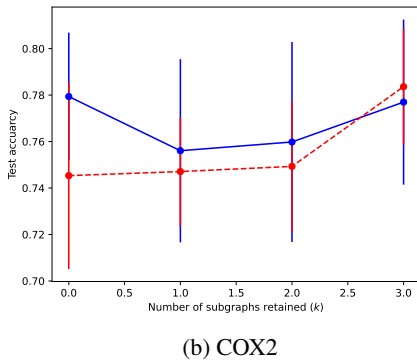

(a) PROTEINS                    (b) COX2

Figure 3: Subgraph ablation study. Using only the subgraphs with the largest $\sigma$ score leads to the best performance compared to choosing random subgraphs during classification.

One could then simply inspect motifs with large $\sigma$ values for potentially explanatory subgraphs and owing to the representation term in the training, all similar subgraphs are also given for free for inspection. Generally, we do not expect that motif-likeness will contain information about graph-level properties (e.g. classification label) for all datasets. However, the large body of motif mining literature contains many examples of settings where motifs do point towards network function (Acosta-Mendoza et al., 2012; Milo et al., 2002; Leontis et al., 2006). We leave further inspection of these subgraphs for future work.

## 5 CONCLUSION

We introduced a framing of the approximate network motif problem in a manner suitable for machine learning models, and propose a first model architecture to address this challenge. The model we propose for this task is able to efficiently recover motifs over baselines in several synthetic data conditions with decoding times on the order of seconds. When placed in a classification setting we show that motif mining has potential as a challenging pre-training step and that the obtained motifs show potential to be naturally used as interpretable feature extractors. Although the inductive nature of our model provides a significant performance boost at inference time, the subgraph similarity loss computation is a quadratic of a polynomial time kernel which could limit the sensitivity of the model. Additionally, the decoding phase requires tuning of several parameters which has to be carefully chosen for each application. Further exploration of of efficient subgraph representation functions Bevilacqua et al. (2021) and graph partitioning Bouritsas et al. (2021) models will therefore become relevant topics for motif mining. We also expect that the specific choices of the modules in `MotiFiesta` will be highly domain-specific. Certain datasets will naturally call for different notions of subgraph similarity, and certain datasets will tolerate varying degrees of distortion in their motif occurrences. Future work should also focus in further exploring the impact of these module choices.

## 6 AVAILABILITY

The code of all experiments is made available as a tarball. It will be made available on GitHub after the anonymous phase.

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

# Appendix

The appendix contains additional details regarding dataset preparation, benchmarking, and experiment setup.

## 6.1 DATASET PREPARATION

To test motif mining models on datasets with ground truth motifs we generate synthetic graphs and repeatedly insert motifs to create over-represented subgraphs at known positions. The dataset construction process admits several choices including number and type of motifs to insert, abundance and degree of distortion across motif instances. For a given motif graph $m$ and randomly generated graph $g$, we randomly sample a node from $u \sim \mathcal{N}(g)$ to delete and replace with $m$. Each $g$ is generated using the Erdos-Reiny random graph generator with a connection probability $p = 0.1$. Next, we randomly sample pairs of nodes from $(u, v) \sim \mathcal{N}(g) \times \mathcal{N}(m)$ as the set of edges that connect $m$ to the rest of $g$. In all experiments, the graph containing the motif is chosen to be twice as large as the motif. Distorting a motif according to probability $\epsilon$ is done by iterating over all node pairs $u, v \in \mathcal{V} \times \mathcal{V}$ and with probability $\epsilon$, we create an edge between $u$ and $v$ it does not exist, and delete the edge if it does exist. Table 3 summarizes these parameter choices.

| Variable | Values | Description |
|---|---|---|
| Motif type | barbell, wheel, random, star, clique | Topology of planted motif |
| Motif size | 3, 5, 10, 20 | Number of nodes in planted motif |
| Concentration | 30%, 50%, 100% | Fraction of graphs where motif appears. |
| Number of motifs | 1, 3, 10 | Number of motifs in single dataset. |
| Distortion probability ($\epsilon$) | .01, .02, .05 | Probability of distorting an edge in motif. |

Table 3: Synthetic dataset construction parameters

## 6.2 ADAPTING EDGEPOOL FOR MOTIF MINING

The EdgePool layer introduced in Diehl (2019) computes a new graph $G'$ from an input graph $G = (V, E, \mathbf{X})$ by first computing a score $s_{uv} = \text{MLP}(x_u, x_v) \quad \forall (u, v) \in E$. A greedy algorithm chooses edges with the highest scores first and collapses them into a new node we denote as $uv$, and computes a new node embedding $x_{uv} = \text{Pool}(x_u, x_v)$. The Pool operation is typically a sum pool but can be any differentiable operation. Contraction is stopped when no edges that connect previously pooled nodes are left. Nodes that do not belong to a contracted edge are assigned to a new node in $G'$. That is, after the pooling step, every node in $G$ is assigned to exactly one node in $G'$ and contracted pairs of nodes are mapped to the same node in $G'$. In `MotiFiesta` we wish to allow for the possibility that some graphs do not give rise to motifs if they do not fit the concentration criteria. For this reason we replace the greedy contraction algorithm with a random sampling. Each edge is assigned a probability using a Sigmoid layer and the new node set is computed by iterating through each edge and adding it to the pooled edges with probability proportional to $\sigma(u, v)$. As such, our version of coarsening is stochastic. Additionally, EdgePool computes joint embeddings $x_{uv}$ only after deciding to contract an edge. Instead we wish for the model to make the pooling decision as a function of the *joint* subgraph and thus in our modification, $s(u, v) = \text{Sigmoid}(\text{Pool}(x_u, x_v))$, for which joint embeddings are computed before the merging step.

## 6.3 MOTIFIESTA ARCHITECTURE

The models built for main results follow the hyperparameters choices outlined in Table 4. Once a model is trained, the LSH-based decoding phase admits two choices. The first is the dimensionality of the hash digest which we vary from $\{8, 16, 32\}$ and which controls the collision probability when assigning embeddings to a motif label. Larger hash digests have lower collision probabilities and

are therefore more sensitive to variability in motif structure. The second choice in decoding is the `MotiFiesta` pooling layer to use, for larger motifs we choose higher layers. This is also varied from 2 to 4 in our experiments.

| Parameter | Values |
|---|---|
| Embedding model | MLP, ReLU activation |
| Embedding size | 8 |
| Pooling layers | 4 |
| Scoring model | MLP, Sigmoid activation |
| $\lambda$ | 1 |
| $\beta$ | 1 |

Table 4: Hyperparameter choices for `MotiFiesta`

## 6.4 DECODING ALGORITHM

In Algorithm 3 we describe the procedure of going from an `MotiFiesta` model to a $K$ dimensional labeling containing the top $K$ motifs.

---

**Algorithm 3** *Motif decoding process.* Compute an integer code for all node embeddings to assign each subgraph to a motif ID. We rank motif IDs by the mean pooling score $\sigma$ of its constituent subgraphs keeping only the top $K$.

---

1: **Input:** a batch of graphs $\mathcal{G}$, `MotiFiesta` model $h_\theta$, coarsening level $T$, number of motifs $K$
2: **Output:** node-to-motif assignment matrix $\hat{Y} \in [0, 1]^{|\mathcal{V}(\mathcal{G})| \times K}$
3: $\mathbf{Z}, \mathbf{S} \leftarrow \mathbf{h}_\theta(\mathcal{G}, \mathbf{T})$ collect all node embeddings after $T$ layers and scores
4: $H \leftarrow \text{LocalitySensitiveHash}(Z)$ map each embedding to an integer code.
5: $\mathcal{H} \leftarrow$ set of unique hash codes in $H$.
6: $\hat{Y} \leftarrow \text{OneHot}(H, \mathcal{H})$
7: Let $R(\mathcal{H}; \mathbf{S})$ return the rank of a hash code by descending mean score of all subgraphs with the same hash.
8: **for** $h \in \mathcal{H}$ **do**
9:    **if** $R(h) > K$ **then**
10:       $\hat{Y}[h] \leftarrow \mathbf{0}$
11:    **end if**
12: **end for**
13: **return** $\hat{Y}$

---

## 6.5 OBSERVED RUNTIME

Training time on typical synthetic runs using 1 NVIDIA GeForce GTX 1080 varied between 4-8 hours on the synthetic datasets. We benchmark the decoding time on one of our synthetic datasets of 1000 graphs of 20 nodes each. Results for 30 runs for decoding at 1 to 4 pooling layers are shown in Fig 4. Decoding was performed on a personal laptop using 1.6GHz dual-core Intel Core i5 processor.

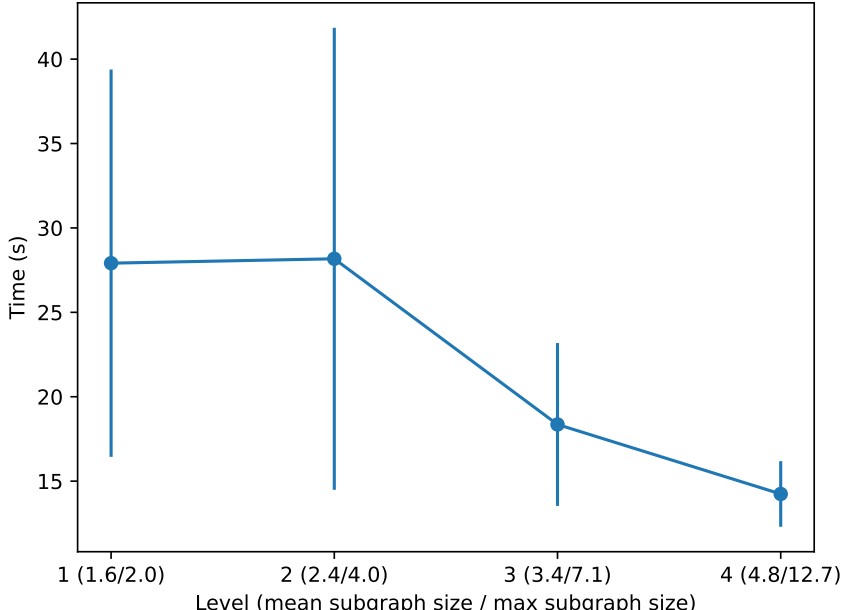

Figure 4: Decoding Runtime. Given a trained model, we decode motifs at each specified pooling layer (1-4, on x-axis) and note the resulting mean subgraph size and largest subgraph found at each layer.

## 6.6 COMPARISON WITH EXACT METHOD

As an external comparison of motif retrieval, we compare with the popular exact motif mining algorithm, MFinder Kashtan et al. (2004a). Since this method enumerates substructures the software crashed after 5 hours on datasets containing motifs of size 5 which is the smallest motif we tested on `MotiFiesta`. Single-motif datasets with size 4 motifs did execute successfully for each of our 5 motif topologies, and 4 distortion levels. Resulting M-Jaccard values are summarized in Table 5. MFinder is unable to retrieve the planted motif for all topologies except for cliques which are almost perfectly recovered.

|  | $\epsilon = 0.00$ | $\epsilon = 0.01$ | $\epsilon = 0.02$ | $\epsilon = 0.05$ |
|---|---|---|---|---|
| **barbell** | 0.00 | 0.04 | 0.08 | 0.22 |
| **clique** | 1.00 | 0.94 | 0.88 | 0.73 |
| **random** | 0.00 | 0.05 | 0.09 | 0.18 |
| **star** | 0.00 | 0.10 | 0.18 | 0.33 |

Table 5: M-Jaccard coefficient using MFinder Kashtan et al. (2004a) on synthetic dataset with motifs of size 4.

Because in MFinder, only isomorphic subgraphs are counted as part of a motif, we observe that when the motif is missed, it is completely missed (M-Jaccard $\rightarrow$ 0). This is particularly noticeable when distortion probability is close to zero, and the opposite trend can be seen for cliques where the algorithm successfully detects the motif it catches all instances and performance decreases with distortion probability. We note that at such small motif sizes, the M-Jaccard that only counts inserted motifs in the motif set can be misleading as randomized graphs can include naturally occurring motif occurrences that the score will not be aware of. Since it is costly to count subgraph occurrences in

general we provide a count of clique occurrences using a clique finder algorithm (CITE). While we manually insert one clique in each graph for 1000 graphs, Table 6 shows that below cliques of size 10, there are actually many more cliques than are inserted. Cliques are enumerated using the NetworkX Hagberg et al. (2008) implementation of Zhang et al. (2005).

| $\epsilon$ | 0.00 | 0.01 | 0.02 | 0.05 |
| **clique size** | | | | |
|---|---|---|---|---|
| **4** | 210000 | 198339 | 870 | 743 |
| **5** | 252000 | 229164 | 0 | 0 |
| **6** | 210000 | 182185 | 0 | 0 |
| **8** | 45000 | 34589 | 0 | 0 |
| **10** | 1000 | 659 | 0 | 0 |

Table 6: True count of clique occurrences in synthetically generated datasets. Each column repeats the count after distorting the motif with probability $\epsilon$.

### 6.7 COMPARISON TO NEURAL METHOD

As an additional external comparison we take SP-Miner Ying et al. (2020), the most similar approach we could find. The SP-Miner model computes the most unnormalized frequent subgraphs and is restricted to isomorphism up to the WL isomorphism test. For this reason the results are not included as part of the main results. Nevertheless the results highlight the importance of modeling subgraph similarity as well as the use of significance testing for motifs. M-Jaccard scores for analogous settings to Table 1 are show in Table 7.

| $\epsilon$ | 0.00 | 0.01 | 0.02 | 0.05 |
| **dataset** | | | | |
|---|---|---|---|---|
| **barbell** | 0.008 | 0.008 | 0.010 | 0.011 |
| **clique** | 0.018 | 0.012 | 0.009 | 0.008 |
| **random** | 0.011 | 0.008 | 0.007 | 0.007 |
| **star** | 0.014 | 0.014 | 0.012 | 0.009 |

Table 7: M-Jaccard results for SP-Miner Ying et al. (2020) neural frequent subgraph miner.

## 7 ANALYSIS AND VISUALIZATION OF MERGING SCORES

After training `MotiFiesta` on a dataset, we compute the M-Jaccard coefficient both on the original data distribution, as well as on datasets with increasing degrees of distortion applied to the motif subgraphs. Before applying the decoding step, however we can already assess the quality of the subgraph scoring layer. Since by construction (see Appendix 6.1) we know which nodes belong to motifs and which do not, we should expect that measuring the merging score assigned by the model in motif nodes should be significantly larger than for non-motif nodes. In Figure 5b see a clear separation between the two distributions, indicating that the model is able to assign proper scores to enriched subgraphs. As a visual aid, we show the pooling process on an example graph in Figure 5a.

## 8 M-JACCARD CONTINUED

In Table 8 we include the results of experimenting on datasets with varying number of motifs (3 motifs, 5 motifs) as well as different concentrations (sparsity 0.3, 0.5, 1) where the sparsity represents the probability that a given graph will contain a motif instance.

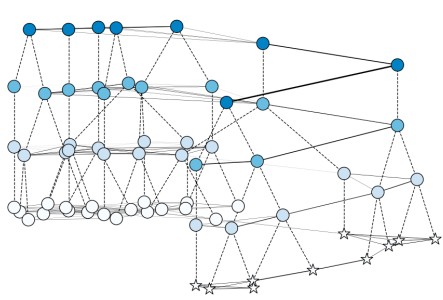
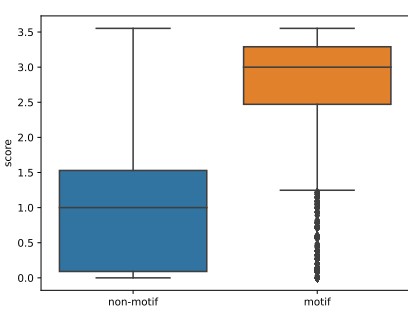

(a) Illustration of edge pooling execution.    (b) Pooling score distribution.

Figure 5: (a) input graph is at the bottom with white nodes, and motif nodes drawn with a star shape. Pooled nodes are connected to the collapsed node by a dotted line. Illustration shows stacked 3 pooling layers. (b) distribution of edge scores assigned to nodes within a known motif versus those outside of the motif subgraph.

| | $\epsilon = 0$ | $\epsilon = 0.01$ | $\epsilon = 0.02$ | $\epsilon = 0.05$ |
|---|---|---|---|---|
| **3 motifs** | $0.19 \pm 0.01$ ($0.18 \pm 0.01$) | $0.29 \pm 0.05$ ($0.18 \pm 0.01$) | $0.24 \pm 0.06$ ($0.17 \pm 0.00$) | $0.29 \pm 0.06$ ($0.19 \pm 0.01$) |
| **5 motifs** | $0.11 \pm 0.00$ ($0.11 \pm 0.00$) | $0.11 \pm 0.00$ ($0.11 \pm 0.00$) | $0.11 \pm 0.00$ ($0.11 \pm 0.00$) | $0.11 \pm 0.00$ ($0.11 \pm 0.00$) |
| **sparse-0.30** | $0.36 \pm 0.00$ ($0.33 \pm 0.02$) | $0.36 \pm 0.01$ ($0.35 \pm 0.00$) | $0.35 \pm 0.01$ ($0.34 \pm 0.01$) | $0.35 \pm 0.00$ ($0.34 \pm 0.00$) |
| **sparse-0.50** | $0.40 \pm 0.03$ ($0.34 \pm 0.00$) | $0.43 \pm 0.02$ ($0.34 \pm 0.00$) | $0.40 \pm 0.04$ ($0.34 \pm 0.00$) | $0.43 \pm 0.05$ ($0.34 \pm 0.01$) |
| **sparse-1.00** | $0.40 \pm 0.02$ ($0.34 \pm 0.00$) | $0.42 \pm 0.03$ ($0.32 \pm 0.02$) | $0.41 \pm 0.02$ ($0.34 \pm 0.00$) | $0.42 \pm 0.04$ ($0.34 \pm 0.00$) |

Table 8: Continuation of M-Jaccard score under various synthetic motif mining conditions. Each row contains the M-Jaccard score when testing on motif datasets with each edge of the motifs distorted with probability $\epsilon$. Unless specified otherwise, motif instances are of size 10 nodes. The values in small type and parentheses are the scores obtained by a dummy model that assigns all edges a merging probability of $0.5$.

