# OpenReview forum: "Learning to mine approximate network motifs"
_ICLR.cc/2023/Conference — Submitted to ICLR 2023_

### Official Review · Reviewer_HL23 · 2022-10-20

**Confidence:** 3
**Correctness:** 3
**Technical Novelty And Significance:** 4
**Empirical Novelty And Significance:** 4
**Recommendation:** 5

**Clarity, Quality, Novelty And Reproducibility:**

The clarity is good, even though can be improved. The problem is clearly stated, however, the proposed model is complicated for first-time readers. Try to improve the explanation of the model. For example, you could explain more based on Figure 2. Also, expand your explanation about Algorithm 1, most of the work is given to the reader. Note, there is no algorithm 2 in the paper.

Novelty is very good, I do not remember a formulation of motif occurrences as machine learning problem. This is very interesting and can give a different point of view on this type of problem. Also, the proposed model is the first to solve this type of problem.

Unfortunately, the quality of the evaluation must be improved. Please see my previous comments.

**Strength And Weaknesses:**

The main strength of the paper is the formulation of motif occurrences as a machine learning problem. The formulation is simple and it can open a new door for motif estimation. Another important contribution is the new proposed model. Both elements are interesting and publishable, however, the evaluation and time complexity must be improved.

There are several papers that calculate the exact number of motifs (for small networks), and other approximation algorithms that try to estimate motifs. However, none of them are considered in this paper. So, currently, we have a model that can extract large motifs and approximate motifs, but there is no ground truth telling us that it works.

To improve your paper, empirically demonstrated that you are able to extract these motifs from large datasets. Maybe, instead of applying 4 o 5 node motifs reduce it to three and use some known datasets.

Another issue is the time complexity of the model, which is O(n^3\log(n)). This time complexity makes the model unfeasible for medium size networks. Maybe, it is better to have an approximate algorithm able to extract some medium large motifs rather than applied this model.

minor comments:
section 1: "size ,"
subsection 2.1: "ponts"

**Summary Of The Paper:**

The paper proposes the task of identifying motif occurrences as a machine learning problem. Then, it proposes a new model called MotiFiesta. MotiFiesta is separated into two parts, a subgraph embedder, and a subgraph density estimator. Unfortunately, the results are inconclusive, given that the model is not compared against any baselines or applied to any network with known ground truth motifs (even though this is mentioned in subsection 6.6 of the appendix).

**Summary Of The Review:**

The paper proposes a new task a novel model. So, even though the clarity could be improved, the main problem of the paper is the evaluation process, where no baselines are considered. I understand that there are no baselines (considering that this is a new problem), but other motif algorithms can be used to compare the final results.

---

### Official Review · Reviewer_4vV6 · 2022-10-26

**Confidence:** 3
**Correctness:** 3
**Technical Novelty And Significance:** 2
**Empirical Novelty And Significance:** 2
**Recommendation:** 5

**Clarity, Quality, Novelty And Reproducibility:**

This work has originality in its proposed graph mining algorithm. Yet, the quality and clarity of the work could be better improved:

- The experimental section misses comparison with other existing motif mining techniques.
- The presentation of the work could be improved. Currently, the paper is hard to understand:
- The presentation of the experimental results is not self-contained, which may cause confusion to the readers.
- The empirical improvements of using motif mining to downstream tasks look incremental. For example, the vanilla GIN model could outperform MotiFiesta in the IMDB-Binary dataset.


**Strength And Weaknesses:**

Strength:
-

[+] The proposed motif mining method has some novelty.

[+] The work also empirically demonstrates that ELF Distillation performs better than a variety of strong baselines on a suite of minigrid environments.

Weakness:
-

[-] The experimental section misses comparison with other existing motif mining techniques.

[-] The presentation of the work could be improved. Currently, the paper is hard to understand:

- It is difficult to see how different sections and their subsections are connected. It would be good to add transition paragraphs/sentences in front of each (sub)section to explain how they are connected to the earlier content.

- The presentation could be better improved with better usage of terminology. For example, what is the difference between graph coarsening and motif mining? It seems that many terms are used directly without definitions.

-  It would also be much clearer if the paper could provide some concrete examples justifying the choice of these parts used in its proposed graph mining algorithm: search strategy, loss function, scoring function, and so on.


[-] The presentation of the experimental results is not self-contained, which may cause confusion to the readers.

- For example, in Table 1, what does the first column represent? Also, it is hard to see the trends via numeric values (figures are recommended for better visualization of comparison).

[-] The empirical improvements of using motif mining to downstream tasks look incremental in Table 2. For example, the vanilla GIN model could outperform MotiFiesta in the IMDB-Binary dataset.

Minor comment:

- The citation style doesn't look right.

**Summary Of The Paper:**

This work formalized the notion of motif mining as a node-assignment problem in machine learning. It proposed an approximate motif mining algorithm, MotiFiesta, which utilizes the composability of motifs in a differentiable manner by contracting edges. This work then empirically showed that motif mining could also serve as an effective unsupervised pre-training routine and interpretable feature selector in real-world datasets.

**Summary Of The Review:**

This work formalized the notion of motif mining as a node-assignment problem in machine learning. It proposed an approximate motif mining algorithm, MotiFiesta, which utilizes the composability of motifs in a differentiable manner by contracting edges. This work also empirically showed that motif mining could also serve as an effective unsupervised pre-training routine and interpretable feature selector in real-world datasets.

Overall, this work has originality in its proposed graph mining algorithm, but the quality and clarity of the work could be better improved. Therefore, I recommend borderline rejection for this paper.

---

### Official Review · Reviewer_Vrqg · 2022-10-30

**Confidence:** 5
**Correctness:** 3
**Technical Novelty And Significance:** 2
**Empirical Novelty And Significance:** 2
**Recommendation:** 5

**Clarity, Quality, Novelty And Reproducibility:**

## Novelty
Due to the lack of proper citations and positioning with respect to other works, it is hard to assess the novelty of the work precisely. From what I understand, the novelty comes from the unsupervised nature of the framework (in contrast to the papers below).

## Clarity and Quality:
The work has some presentation problems. Many citations are missing, both from motif mining and ML for motif mining, some examples:
a) "Neural Subgraph Isomorphism Counting", Liu et al.
b) "Can graph neural networks count substructures?" Chen et al.
c) "Motivo: fast motif counting via succinct color coding and adaptive sampling" Bressan et al.
d) "Unsupervised Joint k-node Graph Representations with Compositional Energy-Based Models" et. al.
e) "Sequential stratified regeneration: MCMC for large state spaces with an application to subgraph count estimation" Kakodkar et al.
f) "Graph convolutional networks with dual message passing for subgraph isomorphism counting and matching" Liu et al.
And many others...

Here are some examples of clarity problems:

-- First paragraph: The task is actually NP-complete, the survey cited just says "the problem is hard". Correct reference is https://epubs.siam.org/doi/10.1137/0210002
-- 2.1: E ⊆ VxV ( instead of ∈  )
-- Last sentence in paragraph below Equation 2 is hard to understand.
-- Figure 2 is also hard to understand, I was able to grasp the idea more from the text than from it. Maybe try to separate the pipeline from above from the loss figures.
-- Algorithm 1 should be self-contained. Currently, we cannot understand it without getting back to the text all the time in different parts to find what function is being used. If it's too hard to define things inside, please point (hyperlink) to the place it is defined in the text.
--- Section 3.5 is a bit confusing. What's the message here?

## Reproducibility:
The appendix of the paper is very good regarding this. Everything I could check for is described there for reproducibility.

**Strength And Weaknesses:**

## Strength

The main contribution of this work is an unsupervised formulation of the network motif mining problem (although it is not how the authors call it). All previous contributions relied on supervised learning to count/mine subgraphs. Here, the authors acquire the learning signal from the similarity function. The problem approached is quite important and often overlooked in the graph learning community, thus I think it is an important direction to follow.

## Weaknesses

a) There is a fundamental problem (not acknowledged) with motif mining as a node assignment problem: it marginalizes subgraph representations. The main thing to keep in mind here is that subgraphs function as a joint structure, e.g., removing or adding a node can impact drastically whether the others' are in the subgraph or not. Think of having nodes a,b,c,d forming a square and choosing to add a node e that is connected to all of them vs adding a node f that is connected to only one of them. The issue with this marginalization procedure is vastly discussed in [1]. This issue is carried over to MotiFiesta, where coarsening iteratively induces the same problem. I understand it is a strategy to overcome the combinatorial structure of joint representations, but i) it should be clarified and discussed and ii) there are works addressing this via sampling[1] or combinatorial representations[2].
b) The clarity issues with this paper are not so trivial. Paragraphs are too long and math definitions are in-line, which makes it hard to understand the work (see clarity section for more details).
c) The work overlooks a huge part of existing literature in subgraph sampling and neural networks for subgraph count methods. I understand it's not *exactly* what the authors are proposing, but it should be clear what are the differences.
d) The authors do not provide any theory, e.g. convergence guarantees, or extensive experiments. I believe in a conference such as ICLR we should convince readers in at one of this forms.
e) Empirical evaluation is extremely limited. Table 2 has only small datasets, no OGB or Zinc evaluations for instance, and trivial baselines. See [1] in the appendix for an example of such a comparison.  There exists a vast literature in graph pre-training by now. Just to be clear, I understand the authors are not aiming at SOTA necessarily, but it's important to compare against what's more recent out there anyway. Table 1 also contains very small graphs and almost no baseline. Please see some examples of citations and baselines in the Clarity section. Finally, erdos-renyi graphs tend to present very random structure, which facilitates the task of finding an artificially injected motif (in the authors' definition).

[1] Unsupervised Joint k-node Graph Representations with Compositional Energy-Based Models, Cotta et al.
[2] Understanding and extending subgraph gnns by rethinking their symmetries, Frasca et al.

**Summary Of The Paper:**

This work proposes a formulation of the network motif mining problem as a machine learning task. In this task, the authors consider evaluating the top-K approximate motifs (according to some distance function) through an end-to-end learning process. The paper then proposes MotiFiesta, an architecture that leverages EdgePool layers to learn approximate motifs under this new framework. Empirically, the authors evaluate the method when retrieving motifs from synthetic datasets and when serving as a pre-training strategy for graph classification.

**Summary Of The Review:**

I believe the weaknesses and clarity sections provide a good justification for my final score. I would like to make it clear that I like the direction of the paper, I think it's quite relevant. I simply think it needs more work: baselines, citations, some theoretical justification for the marginalization in subgraphs' representations. Also, I think the paper will get substantially better as the authors' improve presentation. I encourage them to resubmit soon once these problems are addressed.

---

### Official Review · Reviewer_oTWr · 2022-11-05

**Confidence:** 3
**Correctness:** 3
**Technical Novelty And Significance:** 2
**Empirical Novelty And Significance:** 3
**Recommendation:** 5

**Clarity, Quality, Novelty And Reproducibility:**

**Clarity:**
The clarity of the paper is adequate for most of it. However, the figures were hard to understand and the caption could be improved to better describe them. The presentation of Table 1 is also poorly done and it is hard to understand how the method is being evaluated or what are the baselines being compared to.

**Quality:**
The quality of the paper has room for improvement. There are relevant literature which is missing and there are many choices in the algorithm that are not well justified and ablation studies are missing.

**Novelty:**
The task proposed in the paper is interesting and relevant. The method proposed is build on existing blocks, but presents a novel solution for a mining task, which is less explored in the context of machine learning and deep learning.

**Reproducibility:**
The paper includes algorithms describing the relevant parts of the method, as well as some experimental details in the appendix. The  code is shared, which facilitates reproducibility. I would recommend including the generated synthetic data (or seeds and procedure to generate them) as well.

**Strength And Weaknesses:**

### Strengths

I find the problem studied in the paper (approximate mining with machine learning) to be very interesting and relevant. By extending mining to approximate motifs, the task becomes more amenable to machine learning techniques, and the paper proposes an interesting way of solving it while still focusing on identifying relevant motifs.

### Weaknesses

While the paper indicates some of the challenges of motif mining in the introduction, it lacks a more in depth discussion of the task and relevant literature, which could better situate and inform the reader by illustrating chaallenges and solutions from the data mining perspective, as well as describing in more detail what makes a machine learning solution both useful and difficult.

As for the machine learning literature, there are relevant works that either aim at applying machine learning to the mining task, or that leverage and discuss usefulness of subgraph structure for feature representation which I'd like to see discussed. Some examples:

- Besta, M., Grob, R., Miglioli, C., Bernold, N., Kwasniewski, G., Gjini, G., ... & Hoefler, T. (2022, August). Motif prediction with graph neural networks. In Proceedings of the 28th ACM SIGKDD Conference on Knowledge Discovery and Data Mining (pp. 35-45).
- Bevilacqua, B., Frasca, F., Lim, D., Srinivasan, B., Cai, C., Balamurugan, G., ... & Maron, H. (2022). Equivariant subgraph aggregation networks. International Conference on Learning Representations.
- Cotta, L., Morris, C., & Ribeiro, B. (2021). Reconstruction for powerful graph representations. Advances in Neural Information Processing Systems, 34, 1713-1726.
- Rossi, R. A., Ahmed, N. K., Koh, E., Kim, S., Rao, A., & Abbasi-Yadkori, Y. (2020, January). A structural graph representation learning framework. In Proceedings of the 13th international conference on web search and data mining (pp. 483-491).

The presentation of the paper could also be improved. Both Figures 1 and 2 were hard to understand. I recommend improving the captions and giving a better description in the text. Also, in Figure 1, it seems as if $h(\cdot)$  can only take value either 0 or 1, whereas the text describe the domain of $h_\theta$ as the continuum $[0, 1]$, which is confusing.

The description of the M-Jaccard coefficient seems a bit out of place. I would recomend moving it closer to the evaluation section, putting it at end of Sec 3 or beginning of Sec 4.

In the Algorithm 1, it is not clear what line 7 means. And the function $\sigma$ is not an argument of the loss functions. How is it updated? How is the sampling step of line 12 taken into account when computing the gradients for $\sigma$?

The results presented in Table 1 are hard to understand. It is not clear what is the comparison being made. What is the baseline? What is being compared?

Finally, there are decisions made in the method which are not well discussed or evaluated, such as the choice of the similarity function, the use if LSH or the choice of the density estimation method. What were the alternatives considered and why are these choiices the best? Was an ablation study performed?

**Summary Of The Paper:**

This paper proposes MotiFiesta, a novel deep-learning model to mine (approximate) motifs. The proposed method combines graph representation learning (via GNNs) and a graph coarsening (pooling) strategy to identify approximate motifs and estimate its frequency by comparison with random graphs (through local randomization). The proposed method is evaluated against random baselines and exact mining algorithms in synthetic datasets. It is also used to build motif-based representations as features for graph classification benchmarks on real world datasets.


**Summary Of The Review:**

The paper addresses an interesting, relevant and difficult problem (graph mining with machine learning). The proposed solution is based on existing building blocks, but their use in solving a motif mining problem is interesting and novel. The relaxation of the problem into mining approximate (but still relevant) motifs is also interesting. However, there are pieces of related work missing, some lack of clarity in parts of the paper and a somewhat weak experimental setup. As such, I don't think the paper crosses the threshold for acceptance.

---

> ### Author Response · Authors · 2022-11-19
> **Response to initial review**
>
> **Strengths**
>
> > I find the problem studied in the paper (approximate mining with machine learning) to be very interesting and relevant. By extending mining to approximate motifs, the task becomes more amenable to machine learning techniques, and the paper proposes an interesting way of solving it while still focusing on identifying relevant motifs.
>
> We are glad to hear this.
>
> **Weaknesses**
>
> > While the paper indicates some of the challenges of motif mining in the introduction, it lacks a more in depth discussion of the task and relevant literature, which could better situate and inform the reader by illustrating chaallenges and solutions from the data mining perspective, as well as describing in more detail what makes a machine learning solution both useful and difficult.
>
> This is a valuable point as the intention of this work is indeed to motivate the idea that machine learning solutions are particularly well-situated for modern motif mining challenges.
> In particular, we were interested in the natural ability of neural representation learning models for efficiently encoding structural proximity and density information which are core subproblems in motif mining.
> We also newly include a mention that the challenge these methods bring is of course the loss of convergence and exactness guarantees that is common in deep learning methods.
> We have expanded the introduction to convey this message.
>
> > As for the machine learning literature, there are relevant works that either aim at applying machine learning to the mining task, or that leverage and discuss usefulness of subgraph structure for feature representation which I'd like to see discussed. Some examples:
>
> > Besta, M., Grob, R., Miglioli, C., Bernold, N., Kwasniewski, G., Gjini, G., ... & Hoefler, T. (2022, August). Motif prediction with graph neural networks. In Proceedings of the 28th ACM SIGKDD Conference on Knowledge Discovery and Data Mining (pp. 35-45).
>
>
> > Bevilacqua, B., Frasca, F., Lim, D., Srinivasan, B., Cai, C., Balamurugan, G., ... & Maron, H. (2022). Equivariant subgraph aggregation networks. International Conference on Learning Representations.
>
>
> > Cotta, L., Morris, C., & Ribeiro, B. (2021). Reconstruction for powerful graph representations. Advances in Neural Information Processing Systems, 34, 1713-1726.
>
> > Rossi, R. A., Ahmed, N. K., Koh, E., Kim, S., Rao, A., & Abbasi-Yadkori, Y. (2020, January). A structural graph representation learning framework. In Proceedings of the 13th international conference on web search and data mining (pp. 483-491).
>
> We thank the reviewer for this rich selection of related work. In the updated manuscript we include these works to motivate the use of subgraphs in representing graphs and the importance of motifs in building powerful representations. All of these works rely on either fixed subgraph extraction policies, or on a known library of motifs whereas we propose a method that jointly extracts motifs de novo and builds graph representations. For this reason we believe that these works boost the relevance of our work as a potential enhancement to these architectures.
>
> > The presentation of the paper could also be improved. Both Figures 1 and 2 were hard to understand. I recommend improving the captions and giving a better description in the text. Also, in Figure 1, it seems as if  can only take value either 0 or 1, whereas the text describe the domain of  as the continuum , which is confusing.
>
> Thank you for this, we have removed Figure 1 as  we agree it may mislead the reader and does not present the core methodological contribution and this section was moved to the evaluation section. Instead we have expanded the caption of Figure 2 (now Figure 1).
>
> > The description of the M-Jaccard coefficient seems a bit out of place. I would recommend moving it closer to the evaluation section, putting it at end of Sec 3 or beginning of Sec 4.
>
> Thank you for this suggestion. We have now moved the M-Jaccard to Section 4.1 which indeed relates to the evaluation.
>
> > In the Algorithm 1, it is not clear what line 7 means. And the function  is not an argument of the loss functions. How is it updated? How is the sampling step of line 12 taken into account when computing the gradients for ?
>
> Line 7 denotes the initialization of two models: $\phi$ which computes embeddings for subgraphs, and $\sigma$ which assigns a score to pairs of such subgraphs connected by an edge. We have made this more explicit.
> We have also clarified the invocation of the loss function.

---

> > ### Author Response · Authors · 2022-11-19
> > **continued..**
> >
> > > The results presented in Table 1 are hard to understand. It is not clear what is the comparison being made. What is the baseline? What is being compared?
> >
> > We apologize for the lack of clarity. This is meant to compare a trained motifiesta model which assigns merging scores based on estimated subgraph frequency to a model which assigns a constant probability to all subgraphs of 0.5. The dummy model's performance is given in parentheses.
> > We will be reshaping this table in the final version so that the MotiFiesta performance, and the dummy model in parenthesis are given their own columns.
> >
> > > Finally, there are decisions made in the method which are not well discussed or evaluated, such as the choice of the similarity function, the use if LSH or the choice of the density estimation method. What were the alternatives considered and why are these choiices the best? Was an ablation study performed?
> >
> >
> >
> > We agree that this is an important point, and further exploration could definitely enhance performance.
> > However, we expect these choices to be domain-specific and will reflect the degree to which subgraphs should be considered equivalent (choice of kernel), and the degree to which the motifs should tolerate discrepancies (LSH parameters).
> > The revised manuscript includes an expanded discussion behind these choices however extensive ablations were not performed during the rebuttal period given the time constraints.
> > Namely, we emphasize that our framework is general and many choices such as LSH parameters which affect the degree of approximateness tolearated in the motifs or the subgraph similarity function will be chosen in a domain-specific manner.
> >
> >
> > **Clarity**
> >
> > > The clarity of the paper is adequate for most of it. However, the figures were hard to understand and the caption could be improved to better describe them. The presentation of Table 1 is also poorly done and it is hard to understand how the method is being evaluated or what are the baselines being compared to.
> >
> > We have updated captions, and improved Table 1 to include further baselines and explanation.
> >
> > **Reproducibility**
> >
> > > The paper includes algorithms describing the relevant parts of the method, as well as some experimental details in the appendix. The code is shared, which facilitates reproducibility. I would recommend including the generated synthetic data (or seeds and procedure to generate them) as well.
> >
> >
> > The submitted repository contains seeds for generation, and a script to generate all datasets from scratch (`build_data.py`) in a reproducible manner. We have added this explanation to the README.
> >
> > **Summary Of The Review**
> >
> > > The paper addresses an interesting, relevant and difficult problem (graph mining with machine learning). The proposed solution is based on existing building blocks, but their use in solving a motif mining problem is interesting and novel. The relaxation of the problem into mining approximate (but still relevant) motifs is also interesting. However, there are pieces of related work missing, some lack of clarity in parts of the paper and a somewhat weak experimental setup. As such, I don't think the paper crosses the threshold for acceptance.
> >
> > We hope that the expanded discussion of related work and the clarification of our experimental results combined with the reveiwer's acknowledgement that this is an important problem for the representation learning community addresses the reviewer's concerns.

---

### Decision · Program_Chairs · 2023-01-20

**Decision:**

Reject

**Justification For Why Not Higher Score:**

There are too many small / larger issues, including baselines and issues with the writing. It is even unclear why an ML method to mine approximate motifs is needed.

**Justification For Why Not Lower Score:**

N/A

**Metareview: Summary, Strengths And Weaknesses:**

This work proposes MotiFiesta, a deep learning model for mining approximate motifs in graphs. The model uses graph representation learning and a graph coarsening strategy to identify approximate motifs and estimate their frequency by comparing them with random graphs. The proposed method is evaluated on synthetic datasets and used to build motif-based representations for graph classification on real-world datasets.

The authors made some good changes during the rebuttal. The work is promising. But reviewers want more in-depth changes than what was allowed during rebuttal. Moreover, it must be very clear why an ML method to mine approximate motifs is needed rather than the existing heuristics.

- One of the reviewers suggested that the paper could be improved by providing an even more in-depth discussion of the challenges and relevant literature in motif mining, as well as discussing relevant machine learning literature on the topic. The presentation of the paper could also be improved, and the description of the M-Jaccard coefficient could be moved to a more appropriate location.
- The work needs to adequately discuss or compare to existing literature on subgraph sampling and neural networks for subgraph count methods. Reviewers suggest that the paper should discuss this issue and consider alternative approaches, such as sampling or combinatorial representations.
- Reviewers also raise questions about the presentation of the method, the long paragraphs, inline math (which is hard to follow), empirical results and the choices made in the method.